# Communication Interface Manager for Improving Performance of Heterogeneous UAV Networks

**DOI:** 10.3390/s21134255

**Published:** 2021-06-22

**Authors:** Laura Michaella Batista Ribeiro, Ivan Müller, Leandro Buss Becker

**Affiliations:** 1Federal Institute of Amazonas, Campus Manaus—Distrito Industrial, Manaus 69075-351, AM, Brazil; l.michaella@posgrad.ufsc.br; 2Department of Electrical Engineering (DELET), Federal University of Rio Grande do Sul (UFRGS), Porto Alegre 90040-060, RS, Brazil; ivan.muller@ufrgs.br; 3Department of Automation and Systems (DAS), Federal University of Santa Catarina (UFSC), Florianopolis 88040-900, SC, Brazil

**Keywords:** wireless networks, UAV, mobility, IEEE standards, message reliability, connection stability

## Abstract

Exchanging messages with stable connections in missions composed of multiple unmanned aerial vehicles (UAV) remains a challenge. The variations in UAV distances from each other, considering their individual trajectories, and the medium dynamic factors are important points to be addressed.In this context, to increase the stability of UAV-to-UAV (U2U) communication with link quality, this paper presents an interface manager (IM) that is capable of improving communication in multi-UAV networks.Given a predefined set of available individual wireless interfaces, the proposed IM dynamically defines the best interface for sending messages based on on-flight conditions sensed and calculated dynamically from the wireless medium. Different simulation scenarios are generated using a complex and realistic experimental setup composed of traditional simulators such as NS-3, Gazebo, and GzUAV. IEEE 802.11n 2.4 GHz and 802.11p 5 GHz interfaces are used for the IM selection. The IM performance is evaluated in terms of metrics from the medium-access-control (MAC) and physical layers, which aim to improve and maintain the connectivity between the UAVs during the mission, and from the application layer, which targets the reliability in the delivery of messages. The obtained results show that compared with the cases where a single interface is used, the proposed IM is able to increase the network throughput and presents the best proportion of transmitted and received packets, reception power (−60 dBm to −75 dBm), and loss (−80 dB to −85 dB), resulting in a more efficient and stable network connections.

## 1. Introduction

The application of unmanned aerial vehicles (UAV) is proving to be extremely useful in a variety of areas, from agriculture to military missions [1,2]. The need to exchange messages appears in many of these applications, either in the form of UAV-to-UAV (U2U) or UAV-to-base (U2B) communications. This allows remote access to difficult areas, where video transmission and additional measurement data traffic are needed, such as in applications for border and cave monitoring, natural disaster monitoring, search and rescue (SAR) missions, and others [3].

Flying ad hoc networks (FANET) is a denomination used to represent an ad hoc network among UAVs. It promotes network scalability, given the capacity to add and remove UAVs from the network. It also aims to ensure network survivability since communication does not rely in a single node (UAV). However, several challenges still prevent the complete establishment of FANETs [4,5], for example, the demand for reliable wireless communications to allow sending and receiving messages with lower risk of loss. The main issues are maintaining reliable connections between high-mobility nodes, signal quality, low delay, and a high message delivery rate between nodes. These metrics are of utmost importance for guaranteeing the quality of service in systems involving mobile UAV networks. UAV interaction approaches (e.g., [6]) denote a fixed point *P* to which the UAVs should fly in order to meet each other, to fly within communication range, so that they can share and update data to perform their tasks. Other approaches aimed at buffering and exchanging messages only during the occurrence of events-of-interest [7] or, alternatively, within predefined time intervals [8,9].

These approaches consider pre-determined activities before the mission, so message exchange is severely affected in the absence of a communication signal (or when the signal quality is poor). In these works, dynamic evaluations are lacking of the communication spectrum to which each UAV node is submitted. Other researchers applied the repositioning of nodes in the occurrence of a detected target [10,11]. For example, in [11], when an UAV from the fleet detects a target, it sends messages to the other UAVs, requesting them to reposition themselves at fixed distance intervals from it. However, other application scenarios cannot cope with trajectory modifications, e.g., on SAR missions that have tight timing restrictions [2,12]. The use of different interfaces or communication protocols in the same network (heterogeneous networks) may be an interesting solution to maintain reliable communication links in networks composed of mobile nodes, given the vast number of successful cases in other mobile networks. Examples are the mobile ad hoc networks (MANET) and the vehicular ad hoc networks (VANET) [13,14].

As shown in previous experiments [15], even supposedly similar interfaces such as IEEE 802.11n and 802.11p behave quite differently depending on the medium conditions. These interfaces have been widely used by researchers, who have applied their standard stack layers to develop solutions for improving UAV network reliability and stability in different mission applications [10,11,16]. The 802.11n became popular in the constitution of FANETs because it provides robust MAC and PHY layers that apply multiple input and multiple output (MIMO) technologies with high throughput and larger bandwidth, features that have been extensively explored in this scenario [4,5]. Another important factor is that IEEE 802.11n is commonly found in UAVs classified as common commercial devices (low cost). IEEE 802.11p 5 GHz was developed for high-mobility networks or dynamic time-varying environments (i.e., networks composed of vehicles [14], underwater nodes [17], and robot nodes [18]).

Given this context, this paper presents a communication interface manager (IM) for missions involving multiple UAVs. The proposed IM is situated above the network link layer and contains a heuristic that decides in real-time between two or more wireless communication interfaces. It considers up-to-date monitoring data that represent the current state of the UAVs’ communication links within a given environment. Currently, the heuristic accounts for parameters, such as the number of bytes received, number of bytes lost, throughput, received signal strength indication (RSSI), and signal to noise ratio (SNR). In its current version, the proposed IM is implemented using two wireless interfaces: IEEE 802.11n 2.4 GHz and IEEE 802.11p 5 GHz. The following characteristics summarize the advantages of adopting the proposed IM for multi-UAV communication: (i) provides more stable communication links, (ii) enhances communication connectivity, and (iii) increases message delivery levels.

The proposed IM was validated with simulations conducted using a realistic (and complex) simulation setup based on the NS-3 network simulator, connected to the Gazebo multi-UAV simulator. This allows the UAV missions to be programmed within Gazebo, and leave all communication aspects to be processed within NS-3. Experimental scenarios involving different numbers of UAVs, flying at different speeds, traveling different distances and trajectories were adopted for analyzing the performance of the communication interfaces applied homogeneously (single interface) and heterogeneously (using the proposed IM).

The remainder of this paper is organized as follows: Relevant related works that focused on homogeneous and heterogeneous communication between multiple UAVs are described in Section 2. Section 3 presents the architecture of the proposed IM. Section 4 describes the IM abstract model and the adopted heuristic. In Section 5, the experimentation setup and the created application scenarios are presented. Section 6 highlights the obtained results. Finally, Section 7 presents our conclusions and the future work directions.

## 2. Related Works

Developing a communication architecture for mobile nodes is a challenging task that requires verifying behavioral features (pre-existing interference, coexisting networks, fixed or dynamic amount of nodes), medium access control (MAC) protocol (DCF, EDCA, PCF), the type of network management (decentralized, centralized), and applications-specific traffic (voice, video, data, control signal). Thus, the selection of the communication standard and/or the communication interface to be used should consider the specificity of the networks, and the characteristics in terms of delay, throughput, and link quality variations at a high mobility scenario [3].

Some research initiatives concentrated their efforts on the use of a single communication standard (homogeneous systems) applying cognitive radio techniques to serve these networks, for example, reusing the write spaces slots during communication for management [2]. Other approaches suggest the use of more than a unique standard (heterogeneous systems), expanding the possibilities of communication of the network and managing the communication interfaces according to a context, such as the appearance of interference (noise) in the medium [19].

To provide an overview related to the establishment and control of communication networks systems for multiple UAVs, two sets of related works can be categorized: homogeneous and heterogeneous systems. Additional studies related to vehicular ad hoc networks are also included, since the nodes of these networks have similar characteristics to those of UAV networks.

### 2.1. Homogeneous Systems

Among the researchers that applied homogeneous solutions, Sayyed et al. [20] used a double-stack communication architecture applying bidirectional communication to a UAV node, which establishes communication with the wireless sensor network (RSSF) and its sink node using the same radio. In addition to establishing the link, this architecture provides different QoS characteristics, such as best-effort and reliability responsible for maintaining UAV-RSSF and UAV-Sink links, respectively. The results demonstrated that it can maintain two-way communication using a single radio, based on a flexible communication infrastructure, which defines the protocol stack to be applied according to the detected communication, mobile data collector (MDC), or communications with the sink.

Bekmezci et al. [21] proposed a system with more than one UAV to provide real-time communication. The main objective was to demonstrate that with low-cost hardware and devices easily found in the market, it is possible to create FANETs that meet the requirements of real-time applications. The authors employed an Ar.Drone 2.0, a closed architecture commercial flight unit, along with a Raspberry Pi Model B card with two IEEE 802.11n interfaces, one to allow communication with the other UAV and the other to receive streaming video of the Ar.Drone 2.0 flight. However, the authors only performed latency tests using two UAVs, one static and one that performed a fixed trajectory along the signal range of the static UAV, without evaluating the effects of the use of the second interface in this communication, which could have caused variations due to network coexistence.

Yanmaz et al. [10] proposed a validation of a multi-antenna extension to IEEE 802.11a to be used on multi-UAVs networks and showed the impact of UAV’s height and orientation variations on the link quality for single-hop and two-hops networks. The authors conduct outdoor experiments, measuring the communication performance in terms of throughput and link quality. IEEE 802.11a in access point mode (ground station) and IEEE 802.11s extension in mesh point mode were used to provide one-hop communication (from a single UAV to a ground station) and two-hops communication (mesh networking between two UAVs and one ground station), respectively. Experimental results show that stable throughput can be achieved using two-hop networks where all traffic goes through an access point UAV as relay. The authors conclude that the standard mesh protocol will be insufficient for multi-UAV systems if high throughput is necessary to deliver large amounts of sensor data (e.g., in SAR missions) or if stable links are required to support users. The use of one UAV in AP mode in two-hop architecture implies in lower jitter. This highlights the difficulty one can find for defining a communication system that can support different network requirements.

Lei et al. [22] developed a solution employing changes in the MAC layer of the communication standard to IEEE 802.11p, using optimization mechanisms to avoid inefficiency in the prioritization of traffic (idle time) resulting from the use of the enhanced distributed channel access (EDCA) mechanism. EDCA is applied by the IEEE 802.11p standard to prioritize messages in access classes (AC) (voice, video, best effort, and background). The objective is to propose a solution to the problem of reliability in traffic safety messages in vehicular networks. This solution can also be considered by networks composed of UAVs, as it considers the mobility characteristics that also exist in these networks. The proposed mechanism allowed a reduction of 20% in the packet loss rate, improving the network throughput with each addition of new vehicles, presenting performance similar to IEEE 802.11p EDCA without changes for cases with small numbers of vehicles.

In [23], the problem of user-demand-based UAV assignment over geographical areas subject to high traffic demands was investigated. The proposed model is based on a cost function for multiple UAV deployment, using user demand patterns to assign a cost and density function to each area and UAVs. The authors used a reverse neural model based on user-demand patterns to match each UAV to a particular demand zone. The goal was to provide continuous data between macro cells and UEs, using the UAVs to enhance load balancing by forming multiple intermediate links between the macro cell and the small cell UEs.The results showed that using UAVs yielded 37.7% fewer delays in comparison with a network comprising small and macro cells without UAVs. Regarding altitude variation, it was shown that increasing the altitude provides less interference and appropriate LOS, but also introduces more delays. The probability of guaranteeing the SINR for a particular user in a macro cell was closer to one in both cases using the UAV, showing high connectivity during the connectivity time in comparison with existing ground-based wireless networks.

Lastly, an emerging solution relates to the use of satellite-based augmentation systems (SBAS), which promote the integration of UAVs in the National Airspace System (NAS). It allows UAVs to operate harmoniously, close to each other, in conjunction with manned aircraft, occupying the same airspace and using the same air traffic management (ATM) systems and procedures [24]. Thereby, performance-based navigation (PBN) can be applied to UAVs to operate similarly to aircraft that rely on satellite positioning. The base idea is to use the existing message exchange structure from PBN to UAVs, through reference stations using its global navigation satellite system (GNSS) to define which stations are used for communication establishment. SBAS was originally designed and developed for civilian aviation, but now it will be probably possible to also use SBASs in non-aviation applications. Some studies reported good performance in terms of battery consumption, extending its flight time, and improvement in its navigation performance without additional communication channels and receipt of correction messages from a ground control system [25,26].

### 2.2. Heterogenous Systems

Among the works applying heterogeneous interfaces, [27] presented a communication architecture in which UAV nodes communicate over a cellular network and an IEEE 802.11p network. The purpose of the study was to provide a secure positioning algorithm for UAVs swarms, using RSSI beacon frames sent by UAVs as a distance estimate. The authors considered a communication scenario where UAVs use the cellular network (3G or 4G) to send messages to this base and IEEE 802.11p interfaces for U2U communication, operating at 5.8 GHz. Network management is centralized, as only beacon messages are exchanged between UAVs. The results showed that the uncertainty rates in relation to the calculation of the distances between UAVs, generated by RSSI values received by the triangulation of neighboring nodes, is reduced to a distance up to 50 m between them. However, it gradually increases above this distance.

A still-open challenge is the access of multi-UAV networks to remote areas, using the limited range of wireless transmissions that operate in unlicensed bands. In [28], the authors used retransmission and routing algorithms to provide a wider range in wireless transmissions in multi-UAV networks. The system uses a wireless interface that operates at 2.4 GHz for communications between the cluster-leading UAV and its base, and a 5 GHz interface in U2U transmission. Management is centralized, as UAVs advance their positions in accordance with the improvement in the RSSI signal received between the swarm leader and the base. The results demonstrated that it is a scalable solution, extending the reach of communication inserting new UAVs, and improving the UAVs’ network connectivity.

In [29], the authors developed a communication system that divides UAVs into cluster heads and members of the cluster. The cluster head UAVs are equipped with interfaces of LTE and WiFi technologies in access point (AP) mode, and the members of the cluster are equipped only with WiFi interfaces in this mode. Cluster heads can connect to a virtual private network (VPN) server to receive control commands through the LTE interface and receive data from the sensors through the WiFi interfaces present in the cluster members. The authors evaluated the system using bandwidth consumption, latency, and received strength signal (RSS). The results demonstrated that the latency is stable up to a distance of 18 m between the cluster heads and the member clusters with support for real-time video transmissions, presenting rates between 4 and 6 Mbps and RSS above −60 dBm considering pre-existing integrated communication interfaces in UAVs (without hardware extensions).

The experimental performance of commercially quadrocopters communicating via IEEE 802.11a was evaluated [10], comparing the infrastructure of one-hop and two-hop wireless communications. The main purpose of this research was analyzing the network layer versus MAC layer relaying in terms of throughput and link quality. The results showed that with the two-hop networks infrastructure, the standard 802.11a using the routing protocol based on number of hops is insufficient for multi-UAV systems if high throughput is necessary to deliver large amounts of sensor data or if stable links are required to support users.

### 2.3. Discussion

Overall, the above-mentioned studies did not evaluate the entire network conditions along the flights. In the studies that adopted heterogeneous communications, many assumed pre-existing mission conditions, defining offline which interface will be used for a given purpose, reducing the network’s adaptability to unexpected events.

To construct a method to overcome such issues, maximizing packet delivery and improving the quality of transmissions, which are dynamically necessary requirements to establish wireless networks composed of high-mobility nodes [11,30], some stacks of protocols present changes in their medium access classes (MAC). These stacks include reduction in the CTS, RTS, and ACK frame generation interval and decreases the size of the packet header, thus reducing the response time in message delivery. An example of this is the stack of the IEEE 802.11p protocol, which serves for communication between vehicles, and provides the alternative of allowing the transmission of messages within a network range based on the geo-positioning, so that the nodes do not need previous authentication in the network [31,32]. Other alternatives to maintain the reliability of these wireless networks include complementary mechanisms to a communication protocol, which function by managing the assignments of the frequency channels [33], and efficient algorithm systems that optimize the quality of the message exchange and the repositioning schemes of the nodes [34] in order to improve their connection intensity [7,35].

Maintaining stable wireless connections between high-mobility mobile devices, such as UAV networks, is a research field facing several challenges due to the mobility, flexibility, and variant topologies that constitute these networks known as FANETs [3]. Some of these static features include, in most cases, limited flight time (as in natural disasters), resistance to interference (for example, channel overlap), and limits imposed by technology on the extent of signal coverage [2,5].

Some emergent technology solutions, such as the use of SBAS communication [24], software-defined networks for UAV-to-UAV communications [36], and the use of IoT protocols [37] have produced promising results in these communication scenarios. In terms of algorithms, biological algorithms [38] and dynamic relay nodes selections [39] have provided signal gains and continuous connectivity in search and rescue missions. In this context of emergent solutions, the use of heterogeneous networks solutions can also be useful for addressing the problem of maintaining connectivity between these types of nodes, seeking quality and reliability in the delivery of messages [14,40].

The IM presented in this paper can make decisions about which communication interface should be used based on dynamically sensed network metrics. It is not a centralized network architecture, but a decentralized network with an ad hoc topology, so that message exchanges are performed in UAV-to-UAV (U2U) mode.

## 3. Architecture of the Interface Manager

This section presents the architecture of the proposed interface manager (IM), which aims to provide heterogeneous communication in networks composed of multiple UAVs. Figure 1 illustrates the layers of the proposed architecture.

According to [5], each UAV that belongs to a multi-UAV system needs to communicate while meeting some requirements: observing the environment, evaluating their own observations and those received from other UAVs, reasoning from the observations, and acting in an effective way. Thereby, each layer illustrated in Figure 1 meets one of these requirements, as further detailed.

Application: In this layer, the mission of each UAV is defined. This includes the definition of trajectories, their mobility features, their goals, the type of messages that will be transmitted by each UAV, the recipients, and their payload. In addition, other details are defined in this layer such as the frequency of sending, definition of the protocol used for routing, and packet size. This layer can synchronize with other applications, in our case, with the NS-3 network simulator and the Gazebo virtual environment.Processing: This layer is responsible for executing systems, mechanisms, reasoning or decision-making entities, and coordinating systems. More specifically, it is responsible for the number of vehicles (nodes) in the network, the amount of interfaces, analyzing the parameters sensed by the medium, and calculating other parameters or conditions. Such parameters are used as the input of the decision-making process. This layer evaluates the parameters received from the sensing layer composed by its own and other sensing nodes to define the interface that will improve the link. To ensure a fair decision, it is necessary to use parameters common to all employed wireless interfaces.Sensing (PHY): This layer is responsible for sensing medium information and metrics (signal strength, background noise, latency, throughput, packet loss, and quantities of sent and received bytes). In the proposed architecture, periodic beacon frames are transmitted to allow calculation of the desired metrics.

In summary, the sensing layer serves to “listen” to the medium to sense the network parameters of interest. The protocol layer is used to enable each interface to calculate and sense such parameters according to their intrinsic characteristics of frequency, range, bandwidth, and transfer rates. Although one interface is chosen for communication, the other(s) remain in a listening state to maintain dynamic sensing. The processing layer uses the collect data to execute a heuristic, which aims to select the best interface to be used at a given moment. Finally, the application layer defines the UAV mission, which includes the messages to be sent by each UAVs. It also details the UAV mobility characteristics, such speed, acceleration, deceleration, and waypoints. The protocol layer and the sensing layer can be considered as modules that support the addition of other wireless interface cards to the system.

To guarantee the survivability of an ad hoc network, each node needs to be connected with at least one neighbor through a strong link with acceptable data flow. The aim of this method is to establish the reliable exchange of messages with the maximum possible QoS and to perform the maximum number of successful packet deliveries.

## 4. Heuristic and System Model

The main goal of the proposed IM is to improve communication connectivity by providing more stable communication links. As a consequence, it increases the reliability of the delivery of messages between the UAVs. Therefore, its use of a set of conditions that are essentially composed of the following parameters: quantity of bytes received, quantity of bytes lost, network throughput, the received signal strength indication (RSSI), and the signal to noise ratio (SNR). These parameters are used to analyze the strength of a connection established between neighboring nodes that use the same network interface.

A strong connection is defined by the composition of various network conditions dynamically evaluated during the mission according to calculated network metrics and to the sensed metrics, calculated online, during each UAV journey. These conditions define a decision heuristic used by the interface manager to make decisions about which interface will provide a strong a connection.

Equal weights for each metric were adopted, aiming to maintain connectivity and stability in the communication links, given that these metrics have the same importance in this kind of high-mobility wireless networks. So, it is not enough to simply have a larger volume of bytes received, but also how often these bytes are received and with which quality they are received. Otherwise, if, in the presence of high loss and low RSSI, such received bytes could be constituted by trash or truncated messages.

So, when heterogeneous communication is used, a set INT is constituted, formed by all communication interfaces that integrate the system.

The decision conditions must be evaluated for each communication interface $INT_{x}$ based on the last sample interval. The heuristic currently adopted is based on a rewards mechanism, where each interface accumulates points based on its performance through the defined decision conditions. Thus, a given interface $INT_{x}$ increases one every time its decision condition $c_{i}$ is better classified, as shown in (Equation 1),
(1)$$\begin{array}{c} INT_{x}^{c_{i}}=\left\{\begin{array}{cc} 1, & \mathrm{if}\;c_{i}^{x}\;\mathrm{has}\;\mathrm{the}\;\mathrm{best}\;\mathrm{performance},\\ 0, & \mathrm{otherwise}, \end{array}\right. \end{array}$$

$c_{i}^{x}$ stores the value of a network metrics *i* from Table 1, sensed or calculated using an interface *x* in the last messages sent/received by each UAV. Thus, the intensity of a connection is defined by the sum of the points obtained from a comparison of the decision conditions obtained by each interface.

The proposed IM is distributed in a set $X={\left\{x_{k}\right\}}_{k=1}^{\left|X\right|}$ of UAVs that are part of an ad hoc network established for a mission. Each UAV represents a mobile node belonging to the network, which must be able to communicate with each other in order to enable the sending of the execution status of the tasks, sharing goals in mission time.

Here, a decision tree (DT) is used to evaluate which communication interface will provide the strongest connection for sending the next group of messages within the current sample interval. The quality of communication is evaluated based on the communication of a node with its neighbors within the communication range.

Notably the DT is run in a distributed way, i.e., it runs separately within each UAV node. The entire process of deciding which interface will be used occurs based on the data of the wireless medium collected by a determined UAV and on the network metrics sensed in the communications with its neighbors.

Table 1 presents the metrics used in the DT. Each of these metrics is applied as a condition $c_{i}$, as stated in Equation (Equation 2).

The IM captures information either through directly sensing the medium or by calculating each decision condition. Therefore, it uses the last frames received by the UAVs, verified at each sample interval $N=n_{0},n_{1},n_{2},\ldots ,n_{n}$ defined for a new metric sample. Thus, the value of a condition $c_{i}$ used in the decision heuristic is the average of the values obtained in the last sensed or calculated samples $\sum_{n-1}^{n}c_{n}^{i}$ for each interface belonging to the INT set.
(2)$$INT_{x}^{c_{i}}=\frac{\sum_{n-1}^{n}\;c_{n}^{i}}{\left(n+1\right)},\forall n\in N$$

Each condition in Table 1 represents a level from the decision tree presented in Figure 2. It reaches the target state (final decision) after checking all the decision conditions, comparing the interfaces that compose the INT set. For cases where the DT does not make any decision for an interface, or at the beginning of the mission, at least one sample interval is considered of the operating samples for each interface. This prevents having no values (null) in a given sample of Equation (Equation 2). So, for example, if three interfaces are used, the first three samples correspond to the exchange of messages between the UAVs, alternating using one of the interfaces, without implying the IM decisions.

As such, the DT is established, constituting a heuristic based on the accumulation of points for each interface $INT_{x}$, with the decision of each level by the interface that presented the best performance in the condition that composes the level. The target state consists of the interface with the best performance in a larger number of conditions featuring a stronger connection.

The target state of the DT is reached from a non-complete binary subtree, since only one interface is chosen for the next message flow.

Therefore, the definition of an interface is an optimal local choice based on the decision conditions evaluated along the levels of the tree,
(3)$$\begin{array}{c} INT_{x}=\sum_{h}^{H}r_{h}^{x},\;where\;r_{h}=1,\forall h\in H\\ x_{k}=INT_{x}\Rightarrow INT_{x}>INT. \end{array}$$

The DT score is composed of the points added by each interface $\sum_{h}^{H}r_{h}^{x}$, where $r_{h}^{x}$ is the accumulative variable that stores the sum of points of an interface along the levels *h* of the set of levels that comprise a tree *H*, in which the interface with the highest sum of points $INT_{x}>INT$ is chosen by a UAV $x_{k}$ for the next sending of messages $x_{k}=INT_{x}$. The other interfaces (not chosen) stay in a state of listening to receive the messages sent by other UAVs that are using other interfaces.

The algorithm worst case complexity is Olog(n), where *n* is the number of tree levels. The number of decision conditions compose the height of tree. It happens that, for a fixed number of conditions like in our case, the system presents a linear complexity *n*.

As such, the algorithm can be implemented with low computational costs. The single requirement is allowing parallel readings of the sensed conditions by the set of interfaces applied, enabling the heterogeneous sensing of the environment. The aggregation of sensing samples from the medium, from the different interfaces applied by interface manager, must be stored in buffering schemes so that the sample intervals are the same.

## 5. Experimental Setup

To validate the proposed IM, a complex and realistic simulation environment suitable for analyzing multi-UAV communication characteristics was assembled. It was composed of three different simulation tools:NS-3: This classic network simulator is used to compose virtual communication nodes assigning interfaces (PHY), standards (MAC), routing protocols, general settings (channel, frequency, and propagation models), loss and fading models, IP address, and other networks settings needed to deploy wireless simulation environments [15,40]. Version 3.30.1 was used.GzUAV: The Gazebo-based framework for ArduCopter multi-UAV simulation allows running multiple-UAV simulations in Gazebo [41]. The set of programs includes a tool called GzUAVChannel, which creates instances of virtual UAVs in Gazebo, keeping the simulation clock strictly synchronized with NS-3 and ArduCopter (allows simulated UAVs to be ready to fly), compounding more realistic wireless network simulations. ROS topics are applied by the framework that integrates the Mavlink protocol commands and simulation clocks in order to compose virtual communication channels for more than one instance of ArduCopter. Thus, each channel is defined by a UAV with different SSIDs, ports, and addresses, generating multiple instances to be connected in Gazebo. This framework also allows writing programs using the DroneKit platform.Gazebo (robot simulation): This tool is used to generate 3D simulations, allowing the creation of realistic scenarios where robots, or populations of robots, perform their missions (robots are UAV in the present study). This allows the testing of algorithm performance and the validation of models. Gazebo allows the definition of real coordinates for UAVs trajectories (waypoints) and realistic experimentation scenarios. Other useful features include: speed definition, UAV flight mode, UAV model, and distance–altitude calculation. The experiments presented in this paper use the features of the Iris UAV RTF Quadcopter.

Using NS-3 and Gazebo with GzUAV integration allowed us to create vehicles with high-mobility features and vast wireless communication possibilities. Figure 3 illustrates the interaction among the adopted tools, as further detailed below.

We considered 17 interaction steps in communication flow, as follows: 1. Setup: This block comprises all the definitions of the nodes and the general network configurations, such as number of nodes, available routing protocols, MAC and PHY configurations, the types of messages (traffic), the transport protocol used, IP addresses, network topology, packet transmission rates, and evaluation metrics. This setup block applies one or more interfaces in nodes and provides all parameters or conditions used in the interface manager (IM) decisions block; 2. The NS-3 inserts in a node the interfaces defined in the set of interfaces INT, using the Setup block features; 3. GzUAV integrates this node in an UAV, which has the trajectory, speed, virtual address, and waypoints; 4. Gazebo inserts the UAV node in a simulation environment, executing its mobility model; 5. Gazebo sends the UAV coordinates and current time to GzUAV, which synchronizes with the NS-3 simulation time and calculates the node position; 6. The IM starts the sensing and spreading of beacon data frames using MAC and PHY interfaces; 7. GzUAV applies this behavior while the UAV is moving; 8. Gazebo sends the current UAV position and execution time; 9. GzUAV receives these data and attaches them to the node to correlate the sensed and calculate conditions with the current position of a node and the simulation time; 10. The IM receives the data and defines an interface for the next round of message transmission (messages produced by the composing message block); 11. The composing message block defines the access classes of messages, data rate, packets sizes, payload, and sender–receiver nodes; 12. The IM sends this message using the interface chosen from the INT set of interfaces; 13. GzUAV integrates with the UAV during the flight (in movement); 14. Gazebo sends the current position, the current execution time, and its distance from other UAVs and from the waypoints; 15. GzUAV synchronizes with the NS-3 simulation time according to distance received and the current position of the node; 16. A status of message delivery is received based on the last steps, using any of the interfaces; 17. Finally, the evaluation metrics are analyzed by the IM and the next messages are sent using the allocated interface.

### Experimentation Scenarios

For the experimentation scene, we adopted the soccer field at the main Campus of the Federal University of Santa Catarina (UFSC), in Florianopolis, Brazil (UFSC’s soccer field GPS coordinate is −27.603916 −48.518370). Three experimentation scenarios were used in the proposed evaluation, containing 3, 5, and 8 UAV nodes. Each scenario depicted straight lines that represented the path to be covered by each UAV. The starting point (0x, 0y), or base point, is the point of origin (take-off) and the destination (landing) common to all UAVs (in reality, there was a few meters’ difference between the two for safety reasons). Each UAV had one or two intermediate waypoints, which represented places where the UAV supposedly performs the mission before returning to the base. The adopted safety-distance was 1.5 m for scenario-1, 2.0 m for scenario-2, and 8.0 m for scenario-3, allowing them to start their missions at the same time, avoiding collisions during departure and arrival. Each vehicle had different maximum speeds to be used along the trajectory with an acceleration of 0.8 m/s${}^{2}$, deceleration of 4.5 m/s${}^{2}$, and an imperfection of 0.5% in the execution of the vehicle’s trajectory to and from the intermediate waypoint. The total distance traveled and the maximum speeds for each vehicle are shown in the legends included in Figure 4, Figure 5 and Figure 6.

The traffic generator is composed of an on–off pattern. The duration of each of these states is determined by random time variables in intervals of up to 1 ns. During the off state, no traffic is generated. During the on state, constant bit rate (CBR) traffic is generated with packet sizes of 1472 Bytes. The interface data rate is 50 Mb/s divided by the number of nodes of scenario, due this being the nominal rate from 802.11 radios. For a fair comparison, this data rate was defined in all the evaluated interfaces, employed either homogeneously or heterogeneously. Using the nominal rate of 50 Mb/s from the interface and the defined CBR traffic, it was possible to establish a theoretical message flow limit. As the individual nodes have the theoretic maximum flow rate of 16.67, 10.00, and 6.25 Mbps for the 3, 5, and 8 nodes experiments, respectively, it would be possible to send 1735.11, 1041.66, and 651.04 samples of GPS coordinates, respectively, which have 9600 bps of resolution. If analyzing this traffic as a video streaming with 480p resolution (640 × 420) using a minimum bitrate of 500,000 bps of a video in MP4 format, it would be possible to send this network at 33.33, 20, and 12.5 fps in the 3-, 5-, and 8-node scenarios, respectively. All developed experiments are let available for download at Github [42].

Scenario-1 is illustrated in Figure 4. This scenario had an area of circa 2200 m${}^{2}$, where each UAV had its course of a 215.41 m traveled distance. Each vehicle had its own trajectory and speed: UAV-1, 20 m/s; UAV-2, 30 m/s; and UAV-3, 10 m/s. In this scenario, the UAVs were initially positioned with distances between them up to 1.5 m, as shown in the start waypoint column values inside parentheses in the legend of Figure 4. The total distance course for UAV-1 was 66.25 m, 100.06 m for UAV-2, and 49.10 m for UAV-3. The UAVs flew to the intermediate waypoint and returned to the start waypoint. So, the distance was the same both ways.

Figure 5 presents scenario-2, which used 5 nodes. This scenario covered an area of circa 2500 m${}^{2}$, where the UAVs traveled a total of 583.62 m. The purpose of this scenario was to increase the message traffic volume and to add a larger number of waypoints and UAVs. This was performed in an area within the range limit of the adopted interfaces, IEEE 802.11n 2.4 GHz and IEEE 802.11p 5 GHz, without the need for gateways, relays, roadside units (RSU), or other range-extender devices.

In scenario-2, the total distance course for UAV 1 was 49.94 m with a 10 m/s maximum speed. UAV 2 flied over 187.24 m with a 30 m/s maximum speed; UAV 3 had one waypoint at a 71.14 m distance, flying at a 10 m/s maximum speed; UAV 4 flew 176.45 m to make a return flight to the waypoint; and UAV 5 flew 98.85 m with a 20 m/s maximum speed.

As in scenario-1, the vehicles reached the intermediate waypoint and returned to the starting point. Vehicles designated with more than one waypoint flew at a speed of 30 m/s, vehicles that presented a trajectory of up to 70 m flew at 10 m/s, and the other vehicles flew at 20 m/s.

Scenario-3 is illustrated in Figure 6. The flight area was increased to circa 5000 m${}^{2}$ and more UAVs (8 nodes) were used in this scenario. The total distance traveled was 560.68 m. The aim of this scenario was to include a larger number of UAVs and cover a larger area to allow nodes to reach farther distances in relation to each other in spatial topology using a single waypoint. The goal was to evaluate the performance when the limit of the interfaces range is slightly exceeded, i.e., evaluating the interface manager and the interfaces used separately in situations where the nodes exceed by a few meters the theoretical wireless signal coverage limit.

In all scenarios, the purpose of this variation in the number of waypoints and speeds was to simulate different distances to be covered by different types of UAV, standardizing the longer distances to be covered for faster UAV, as shown in Figure 4, Figure 5 and Figure 6. A complete trajectory of the mission was defined by the start waypoint, intermediate waypoints, and return to the start waypoint.

Table 2 presents the relevant configuration parameters used by all scenarios. The Friis free-space propagation model was used in the simulation as a signal attenuation model, without obstacles in the line-of-sight (LOS). However, to include signal fading effects, the Nakagami statistical shading model was applied, once the nodes were in non-line-of-sight (NLOS). The occurrences of NLOS in these scenarios were caused by the long distances between UAV in parts of the trajectory, where the fading coefficient was applied.

The Nakagami model was employed using a Rayleigh distribution, considering variations in signal strength due to multi-path fading. The frequency channel adopted by Interface A (IEEE 802.11n 2.4 GHz) was composed of the sum of channels 5 (2.432 GHz) and 6 (2.437 GHz), channels of 20 MHz each, over the Industrial Scientific and Medical (ISM) band at 2.4 GHz, allowing a coverage area of 300 m. Interface B was composed of the IEEE 802.11p 5 GHz standard using channel 172 in the frequency of 5.860 GHz with a 10 MHz width band over the 5.9 GHz (5.850–5.925 GHz) frequency spectrum of the Unlicensed National Information Infrastructure (U-NII) band, with a coverage area from 350 m to 1 km (nominal range, according to the standard).

## 6. Obtained Results

All the experiments were performed using both homogeneous and heterogeneous interfaces, that is, using a single interface (IEEE 802.11n or 802.11p) and using the proposed IM (which switches between them). The performance of each set of experiments is compared and discussed in this section.

The experiment duration varied for each scenario: circa 18 min for scenario-1 (three nodes), 90 min for scenario-2 (five nodes), and 120 min for scenario-3 (eight nodes). The long simulation time was used to realistically simulate scenarios that execute in the Gazebo simulator. NS-3 converts real time into simulation time, as it uses the metrics of evaluation in its execution logs in the order of ns. As such, the simulated flights of the experiments had a duration of 140 s (three nodes), 150 s (five nodes), and 160 s (eight nodes). To evaluate performance, the experiment duration considered in the NS-3 simulator was used.

The IM validation occurs at 1 s intervals using the average of the condition decisions from buffers containing at least 10 samples, collected every 100 ms. This avoided instantaneous value fluctuations. The first and last samples were discarded to minimize the effects of the initial and final execution of the networks in the validation of the results.

Notably, the metrics received from the MAC and PHY layer occur in 1 ns intervals (NS-3 logs), being composed of sensed propagation parameters such as reception and transmission power, path loss, and quantity of transmitted and received bytes.

The obtained results are presented in two groups: (1) application evaluation, and (2) MAC and PHY evaluation. The application group targeted analyzed the reliability of delivery of messages. Therefore, we observed typical network performance metrics such as throughput, packet delivery rate (PDR), end-to-end delay, and latency. To calculate such metrics, the following variables were monitored during the experiments: bytes and packets transmitted (TX), bytes and packets received (RX), lost packets, packet delivery delay, transmission and reception time, and the successful messages flow. The results from the application group were compiled offline after the simulation was completed. The MAC and PHY evaluation considered standard wireless network metrics: amount of bytes received (Mb), Rx power (dBm), loss (attenuation coefficient in dB), delay (ms), received signal strength indication (RSSI), and noise (dBm). Notably, the results of the MAC and PHY group supported the analysis of the improvement and maintenance in the connectivity between the UAVs during the mission. The results from MAC and PHY results were collected online during the mission execution.

### 6.1. Application Evaluation

Network throughput is the first of the application metrics to be discussed. This metric was chosen to evaluate the data flow of the nodes and, consequently, the effective exchange of messages during the flight, considering their different adopted speeds (equal to or greater than 10 m/s). The average network throughput was calculated based on the average throughput obtained by the amount of messages received by the nodes in each network experiment, as stated in Equation (Equation 4).
(4)$$Average\;Network\;Throughput=\frac{\sum_{k=1}^{N}\frac{Amount\;of\;Rx\;Bytes\;(k)}{Last\;Time\;Rx\;Packet\;(k)+First\;Time\;Tx\;Packet\;(k)}}{N}$$

This is composed of bytes received in each message exchanged (amount of Rx bytes (*k*), i.e., data flow), considering the last packet reception time and the first packet transmitted of each data flow (*k*), divided by the amount of network traffic (*N*).

Figure 7 shows the average network throughput from each experimental scenario. The IM performed best in all three scenarios. Another aspect of this experiment was that the throughput ratio decreased while the number of nodes increased. This was expected because the CBR data rate was divided by the number of nodes. Thus, the IM presented 2.5 Mbps in comparison with 0.5 Mbps for 802.11p and 1 Mbps for 802.11n for the three-nodes experiment. In the fives-nodes experiment, the IM presented 1.55 Mbps in comparison with 0.25 Mbps and 0.4 Mbps for 802.11p and 802.11n, respectively. Finally, for the eight-nodes experiment, the IM presented 0.62 Mbps against 0.15 Mbps (802.11p) and 0.25 Mbps (802.11n).

Another observation was that the 802.11n interface, in all three scenarios, always produced twice the 802.11p throughput. This highlights the intrinsic difference in the channel bandwidth of these interfaces, 20 MHz for 802.11n and 10 MHz for 802.11p. The 802.11n interface can have a bandwidth up to 40 MHz using two frequency channels. However, to perform a fair comparison, this was reduced to 20 MHZ with the use of only one frequency channel.

Analyzing the three- and eight-node scenarios, with the average throughput obtained by the network using the IM system, in the three-node experiment, it would be possible to send about 260.41 samples of GPS coordinates per second, considering the use of a GPS device with 9600 bps resolution and about five frames per second of a 640 × 420 (480p) resolution video. For the eight-nodes experiment, it would be possible to send 83.33 samples of GPS coordinates per second and about 1.6 frames per second of the same video. We concluded that this amount of GPS samples is favorable for the proposed SAR experimentation scenarios. In the case of the three-node experiment, the resolution and frames obtained from a possible video transmission by an MP4 resolution camera were satisfactory. However, for the eight-node scenario (more sparse scenario), some method of signal relay should be adopted for this video transmission resolution. Improving this will be the focus of future investigation.

Figure 8 presents the amount of packets (*n*) trafficked in the network considering all the network traffic generated by the complete mission execution. The total amount of packets transmitted and received over the network during the three experiments represents the total volume of data trafficked by the network and, implicitly, the amount of packets lost (total of packets transmitted minus the amount of packets received). This metric describes the productivity of the network, that is, the more packets delivered, the more productive the adopted network settings. It allows a higher message recovery rate and, consequently, a higher volume of effective messages. The best proportion was achieved by the IM in all scenarios as shown in Figure 8. This is also shown in the last column (Flow (n)) in Table 3. In all scenarios, the IM presented the best Tx and Rx packets relation with the highest number of packets trafficked, in the order of 3 × $10^{6}$, 4.25 × 10${}^{6}$, and 3.75 × 10${}^{6}$ with 1 × 10${}^{6}$, 5 × 10${}^{5}$, and 4.5 × 10${}^{5}$, for three, five, and eight nodes, respectively. This highlights that the IM produced almost twice the average throughput of both interfaces applied homogeneously.

Another conclusion that can be obtained from Figure 8 is that although the 802.11p interface had the lowest flow rate in all scenarios presented in Figure 7, its packet productivity was similar to that of the 802.11n interface, even considering its lowest nominal bandwidth. This may have occurred because the packet production interval for this technology is shorter than that for 802.11n. The header size of the WAVE protocol requires a 32 µs transmission time to the physical preamble, as opposed to the 96 µs of 802.11n, as stated in [15].

Figure 9 shows the resulting packet delivery ratio (PDR) (see Equation (Equation 5)) from the network packet generated by the U2U communication. This metric relates to the network packet delivery capacity. This capacity refers to the packets ratio delivered originating from the source node in its application layer, CBR, and the number of packages received by the CBR sink at the final destination [43]. It can be observed that the IM produced the best delivery rate in all scenarios, followed by the 802.11p interface. In this case, the IM presented the best PDR rate, 0.977, which was observed in the five-nodes scenario. In this scenario, the existence of intermediate waypoints allowed a significant increase in the amount of packets trafficked (as shown in Figure 8) and, consequently, an increase in the PDR. As the intermediate waypoints were applied to the UAVs, increasing their courses (UAV-2 and UAV-4 had intermediate waypoints), this allowed these nodes to stay longer within the neighbors’ coverage area: UAV-1 and UAV-3 covered UAV-2, while UAV-3 and UAV-5 covered UAV-4. For the three- and eight-node scenarios, the IM presented PDRs of 0.760 and 0.571, respectively, which implies that more messages were exchanged during the flight compared with using the interfaces homogeneously. In practical terms, we concluded that the IM provided 76% (three-nodes scenario), 97% (five-node scenario), and 57% (eight-node scenario) of the effective capacity to exchange messages during the flight. This demonstrates that the switching resulting from the IM decisions between the available interfaces increases the effectiveness of the network in maintaining connectivity.
(5)PDR(%)=Amount of Rx Packets/Amount of Rx Packets+Amount of Loss Packets

A useful observation is that the IM performance in the five-node scenario, which included intermediate waypoints, highlights the idea that the use of relay UAVs or intermediate relay nodes can be beneficial for distances between vehicles greater than 70 m.

Table 3 presents the average end-to-end delay, latency, and flow of messages in all three scenarios. Each message flow was composed of 1472 byte packages, which simulated a typical application message transmitted between UAVs. The end-to-end delay (ms) and the latency (ms) describe the average obtained from all received packets. As can be observed, the IM performed worse in end-to-end delay (highest value) in all scenarios. In terms of latency, the IM was last only in the three-node scenario, but was ranked second in the other scenarios (ahead of 802.11n). The 802.11p applied homogeneously produced the best performance in end-to-end delay and latency for all scenarios; one should recall, however, that this was achieved with a low messages flow. The 802.11p standard has the feature of using short headers with no need for nodes association to communicate in the setting applied, which explains the least delay and latency. Regardless, the IM still performed the best considering the higher amount of successfully delivered messages.

Considering the transmission of 480p video frames referenced in the discussion in Figure 7, the performance produced by IM in terms of average end-to-end delay and latency is satisfactory, as it received a “Very Good” quality reproduction [44]. However, for the three-node experiment, the observed latency was 29.112 ms, which would cause a reduction in quality to “Good”, by reducing the video resolution. The higher number of nodes in the network implies more neighboring nodes generating lower averages in end-to-end delay and latency and a larger amount of message flow, since the messages are being sent in broadcast mode.

### 6.2. MAC and PHY Evaluation

This section highlights the effects of the proposed IM on standard wireless network metrics. The first metric to be discussed is the amount of application-related data (payload) received by each node, as depicted in Figure 10.

The amount of data generated in the three-node scenario was 5096 Mb, 4011 Mb for the five-node scenario, and 4233 Mb for the eight-node scenario. Figure 10 shows the amount of bytes received by the network in 5 s intervals. The IM decisions (interfaces switching) are also presented. These decisions are composed of the aggregated average of decisions of 5 s, presented using labels with the IM curve. The labels represent the aggregate decision composed of blocks of sample intervals, which the IM considered in most decisions. This means that the interface shown in the samples is the one chosen by the IM on most of the nodes in a given time interval.

With respect to the payload transmission in bytes received (Mb), the IM performed the best only in the three-node scenario (Figure 10a). The IM initially chooses the IEEE 802.11n interface, changing to the IEEE 802.11p interface in the intervals of 20, 40, 48, and 65 s to maintain the growing curve of the transmitted bytes. Thus, for the scenario with three nodes, 802.11n provided a higher transmission rate until to the end of the experiment (141 s). When using homogeneous interfaces, data transmission was interrupted before reaching the end of the experiment, reducing the amount of data received.

For the five- and eight-node experiments, the IM predominantly selected 802.11p, only using 802.11n in the initial 15 s. Three factors can explain the reasons for this occurrence: nodes reached longer distances from each other during the route, up to 75 m in the three-node and up to 100 m in the five- and eight-nodes experiments; the sparser layout of routes; and, lastly, the Rx power factor in the decision tree reasoning. As shown in Figure 11, most of the time 802.11p presented an Rx power between −50 and −65 dbm, which indicates a “Good” or “Strong” signal. The volume of bytes transmitted was smaller than in the experiment with three nodes, as 802.11p has a smaller bandwidth compared with 802.11n. Less traffic was generated (CBR data rate). However, this interface achieved lower flow rates, so we were able to notice a stability in the transmission, which represented a continuous growth with less fluctuations, maintaining the connection without interruptions.

Another observation was that in the five- and eight-node scenarios, the UAVs trajectories covered wider areas, allowing UAVs to be more distant from each other at certain points. Theoretically, the IEEE 802.11p on its on-board unit (OBU) version can reach a coverage of up to 1 km, which would justify its preference given its longer communication range.

Although, the IM did not present the best performance in terms of received bytes in the PHY layer in the five- and eight-node experiments, we observed that achieved values for the IM are satisfactory because the IM presented the best performance in terms of effective message delivery at the application level, as shown in Table 3 and Figure 8 and Figure 9. An effective message delivery means that more messages were successfully delivery at the destination, probably because of the linear and continuous behavior of the PHY layer, with suitable Rx power and loss, which allowed a better message recovery performance at the destination.

Figure 11 presents the performance of the average aggregated of Rx power obtained by nodes over the sample time intervals. For experiment, a sharp curve can be observed, equivalent to the samples derived from the average signal power received from the nodes during their trajectories.

For the three-node experiments (Figure 11), where the interface manager decided more often to use the 802.11n interface, we verified that up to a 10 m distance between the nodes, the reception power increased up to −60 dbm. This was also observed when interfaces were applied homogeneously. These intervals included the time of establishment and spread of the signal by the nodes in the spectrum (the nodes become audible). Thus, from a 20 to 50 m distance between the nodes, we verified that the IM presented a performance similar to IEEE 802.11n, maintaining the signal reception to between −60 and −75 dbm, which is considered a good reception quality for the 802.11 standard. However, the 802.11p interface showed better performance, maintaining the signal reception around −65 dbm, which is considered an excellent reception quality, maintaining the link at these levels up to a 70 m distance between the nodes. The IM possibly chose 802.11n more often due to the higher volume of transmitted bytes (Figure 10), lower attenuation received (Figure 12), and lower delay generated (further discussed).

For the five- and eight-node scenarios, the IM performance was similar to that of 802.11p. As shown in Figure 10b,c, the IM decision was predominantly for 802.11p. An important observation here is that the large number of samples obtained caused a denser curve compared with the 802.11p experiment. This means that the IM switching caused a better signal reception on the network in general, allowing maintenance of message exchanges with good reception power with larger distances between nodes: 70 m (five nodes) and 120 m (eight nodes).

The performance of the Rx power presented by the network operating with IM is considered very good, since it maintained the levels of signal reception at levels classified as high reception for networks constituted by the 802.11 protocol; this allowed network traffic over greater distances between nodes and maintained the connection for a longer period of time, as shown in Figure 10 and Figure 12. In SAR missions, the connection time intervals and greater signal reception by the nodes during the flight are desirable features in the constitution of mobile networks composed of such high-mobility nodes.

Figure 12 shows the loss in dB from the network during the UAV trajectories. These experiments correspond to the average value obtained in each aggregated sample, considering the attenuation caused by Nakagami fading and the power received by neighboring nodes.

The loss (dB) for interface manager experiments in the five- and eight-nodes scenarios presented the lowest signal loss indexes, maintaining a rising curve up to about 82–85 dB. This means that 30% of the path signal attenuation was caused by multi-path transmissions and fading effects between Tx and Rx nodes. In Figure 12, for the three-node experiment, the loss obtained by the IM experiments is similar to that of 802.11n, once the IM decided to use this interface more. However, during 0–30 m, the IM alternated, presenting minor losses compared with 802.11n maintaining the losses between 85–100 dB. Additionally, the variable values of loss from 30 m demonstrated that the IM attempted to decrease this rate, impacting the overall network performance.

The thicker curves generated by the five-node and eight-node scenarios represent a larger volume of data trafficked by the nodes, which can be constituted by messages and by beacon frames along their trajectories. An important point is that the UAVs remained able to communicate for a longer time, reaching greater distances. This is more evident using as reference the typical loss of 802.11n systems of 80 dB in the 2.4 GHz band and 86 dB in the 5 GHz band considering a 100 m distance between the source transceiver and the destination transceiver obtained by free space loss attenuation, which implies a −60 dBm reference Rx power [45]. So, the IM and 802.11p performance for the five-node and eight-node scenarios presented a 85 dB loss maximum with an Rx power of −50 to −70 dBm (Figure 11). Theoretically, the Rx power would be −69 dBm, resulting in 16.02 dBm (power applied in the 802.11 transceivers used (Table 2)) minus the 85 dB maximum of loss in Figure 12.

Figure 13 presents the RSSI and noise aggregated samples received by the network during the experiments. Here, the IM presents the lowest variation in RSSI in the three experimental scenarios, which implies more stable connections between the nodes. The RSSI remained in the range of −38 to −34 dBm in the three-node experiment scenario, from −34 to −30 dBm in the the five-node scenario, and from −50 to −48 dBm in the eight-nodes scenario, allowing greater connectivity of the nodes during the flight.

The intensity of the received noise showed low variation throughout the experiments, remaining in the range of −92, −94, and −96 dBm considering the background noise and the noise caused by the co-channel interference between the nodes, since they used the same channel for communication for both interfaces in all experiments (homogeneous and heterogeneous).

The intensity of the connection in levels can be verified from the result obtained by the RSSI network subtracting its average values of −38 dBm (three nodes), −34 dBm (five nodes), and −50 dBm (eight nodes) by the average noise obtained −92, −94, and −96 dBm, respectively. In this case, we obtained −54, −60, and −46 dBm network RSSIs, respectively. So, on scale of 0 to −100 (with 0 meaning the best signal possible and −100 indicating the worst), the IM presented an RSSI value of around 50, which is generally considered to be good enough for most users and online activities. The SNR was calculated from the average values of the RSSI and noise obtained by the network: 0.41 (three nodes), 0.36 (five nodes), and 0.52 (eight nodes) network average signal gain.

Figure 14 shows the message delays considering interval samples of 10 ms. The delay measurements occurred until reaching the maximum distance between nodes. The label containing the IM decisions is shown up to 150,000 samples for the five- and eight-node scenarios. It caused the IM to not change the use of the last chosen interface.

Thus, in the three-node scenarios, the IM presented better performance for delay measurements, with an average 10 ms delay up to 200,000 samples (around a 20 m distance between nodes). After 200,000 samples, we observed a peak delay of 220 ms, which decreased soon after. For delay validation, only the trajectory to the UAV reaching its maximum distance in relation to the starting point was considered.

For the five-node scenario, we observed an IM performance similar to that of 802.11p, increasing the delay after 60,000 samples at 10 ms (approximately 16 m distance between nodes) and starting an ascendant curve after 100,000 samples (approximately 30 m distance between nodes). The 802.11n interface presented better performance, increasing the delay only after 150,000 samples (approximately 35 m distance between nodes). Notably, this scenario contained a larger area to be covered by the UAVs, which due to the existence of secondary waypoints and to the greater number of nodes, more messages are generated, increasing the aggregated delay. Then, some nodes may have already started returning to the starting point while others were still on the way.

For the eight-node scenario, all the experiments presented 12–30 ms delays up to 150,000 samples (approximately 50 m distance between nodes). In this scenario, the UAV trajectories were sparser but in straight lines (only had start and end waypoints) reaching up 100 m distances between nodes. Therefore, after 200,000 samples, an ascendant curve started up to a 80 ms delay, as the nodes reached their maximum distance.

Observed messages delay related to the use of the IM can be considered acceptable for transmitting 480p video sessions and for sending the GPS coordinates mentioned in Section 6.1, and data and telemetry, where such delays generally do not result in a noticeable decay in experience by a user, for example. However, the delays recorded from 200,000 samples in the experiment with three nodes and five nodes would not be acceptable for voice-over WLAN applications, as connections are typically dropped if the delay exceeds 150 ms [45]. So, depending on the application data to be transmitted over the network, the delays obtained in our experiments are considered tolerable.

In general, the IM selected the best interface most of the time due to a proper dynamic analysis of the medium conditions, in accordance with the variations in the distance between the UAVs. For the five- and eight-node scenarios, with increased communication stress due to the larger communication range and amount of data trafficked, although the IM did not produce the best performance in the evaluated metrics overall, it guaranteed more data being transmitted with more-stable, better-quality connections. This is due to its performance in terms of Rx power (Figure 11) and loss (Figure 12). The IM also presented a greater throughput and PDR, and higher volume of packets transmitted, allowing better message flow.

### 6.3. Final Remarks

Recalling the application metrics, we observed that the IM produced more than twice the average throughput of the best-performing interface applied in homogeneously. This performance allows sending a reasonable amount of GPS coordinates per second and enough frames per second of low-resolution video streaming (640 × 420, 480p). This condition is suitable for low-complexity SAR missions of up to 1 km of total area. We noticed, however, that in the eight-node scenario, the adoption of signal relay would be ideal for increased video transmission resolution.

Another relevant aspect relates to the amount of packets trafficked in the network, where the IM also achieved the highest proportion of transmitted and received packets, which resulted in a more efficient network, allowing a larger flow of effective messages. PDR was used to evaluate the packet delivery network capacity, where the IM managed to achieve 0.977% in the five-node scenarios, around 15 times better than using homogeneously applied interfaces. In terms of average end-to-end delay, the IM performed the worst because this metric also includes the time spent before source transmission, and the IM requires time for the spectrum evaluation and computing its decisions. In any case, the IM still presented the best performance considering the higher amount of successfully delivered messages.

In terms of MAC and PHY metrics, we observed that the IM maintained the exchange of messages during longer time intervals compared with interfaces applied separately. This represents more time available for communication between nodes and a larger amount of received data. For the experiments with more than 100 m between vehicles on sparse routes, the IM predominately selected 802.11p for use in most of the nodes. This means that the IM properly selected the safest interface to maintain network connectivity, since 802.11p provided the best signal reception (Rx power and RSSI), maintaining the intensity of noise and signal losses within acceptable levels with an imperceptible delay rate for the type of traffic adopted.

The obtained results also allowed to reinforce the assumption that all adopted criteria for selecting the best interface should be used with similar weights. For example, if increasing the weight of the amount of received bytes, for instance, these bytes could be constituted by trash or truncated messages in the presence of high loss and low RSSI.

Finally, even with the lower flow rates presented by the IM (Figure 10 and Figure 13), using the IM allowed increased stability in the transmissions, with continuous growth and less fluctuations, reducing the traffic interruptions caused by signal losses. This reinforces the assumption that the adopted metrics for selecting the best interface should be used with similar weights. If, for instance, a higher weight were given to the amount of bytes received, the 801.11n interface would have been selected, but there would not be as much successfully delivered messages at the destination nodes as achieved in the conducted experiments.

## 7. Conclusions and Future Work

Collaborative missions or missions with shared tasks that employ wireless multi-UAV networks require higher link stability and special attention to the reliable delivery of messages. An important part of this challenge relates to the wireless interface technology that is used. In this context, in this paper, we proposed a heterogeneous interface manager (IM) that is capable of dynamically defining the best communication interface to be used when more than one option is available. It works by dynamically evaluating parameters collected from the medium. The benefits of the proposed method are that U2U connectivity is improved and message delivery reliability is enhanced.

The current version of the proposed IM works with a heuristic formulated as a decision three. For validating the proposed method, a close-to-reality simulation scenario was constructed with the use of different simulation tools. From obtained results, the proposed IM presented a lower performance fluctuation in the network flow, even with the increase in the number of nodes, presenting good reception of signal by nodes during all missions, with fewer communication interruptions. The IM was able to maintain the link between −60 and −75 dbm, with a distance of up to 100 m between the nodes, with the lowest signal loss indexes (80 dB to 85 dB), and with SNR up to 0.52. Thus, the use of the IM promotes higher link stability, enabling the dynamic selection of interfaces in order to adjust the network to the conditions of the medium and to promote U2U communication with increased reliability. The IM enabled almost twice the sum of the average throughput from both interfaces applied homogeneously.

In future work directions, the design of a new heuristic that considers different classes of communication messages should be addressed. This could, for instance, favor some of the metrics over the others. In Non-delay tolerant networks (NDTN), for example, it could be interesting to assign a higher weight for delay and throughput metrics. In this respect, our previous results in [15] will be considered regarding homogeneous network behavior (802.11n and 802.11p) using different message classes.

## Figures and Tables

**Figure 1 sensors-21-04255-f001:**
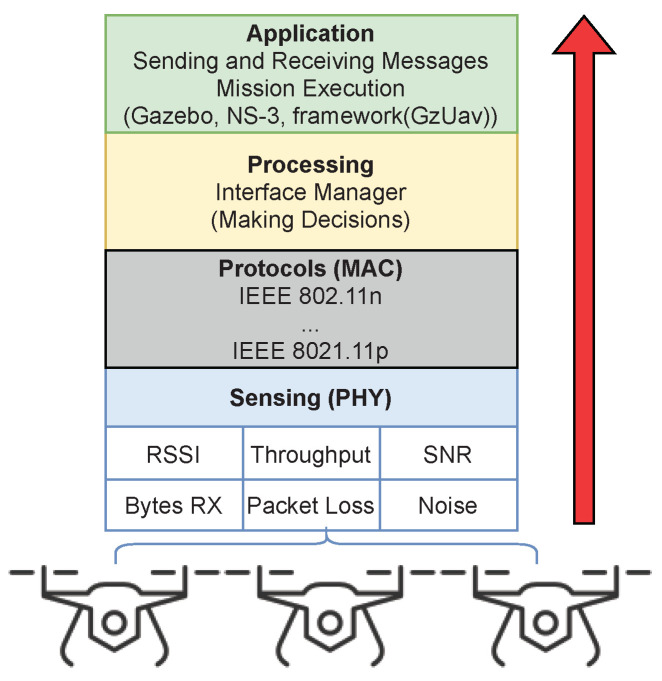
Interface manager (IM) architecture.

**Figure 2 sensors-21-04255-f002:**
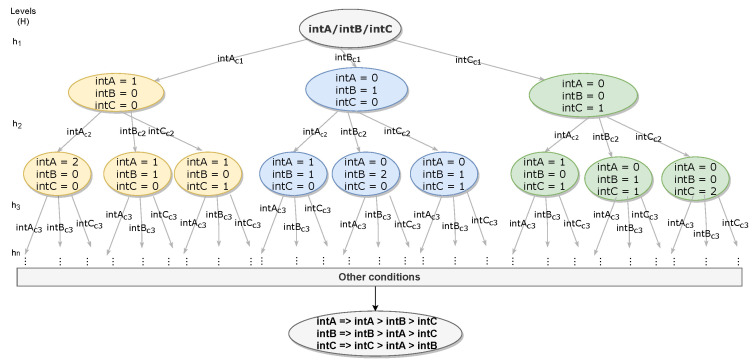
Decision tree used by the interface manager.

**Figure 3 sensors-21-04255-f003:**
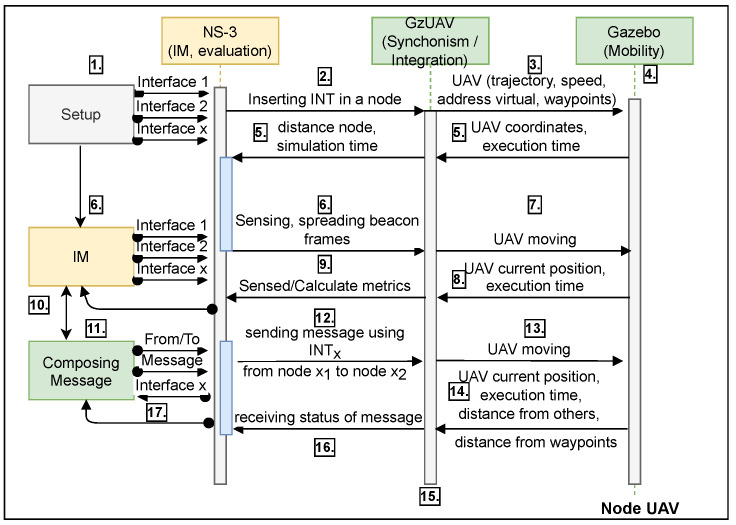
Interaction among the simulation components.

**Figure 4 sensors-21-04255-f004:**
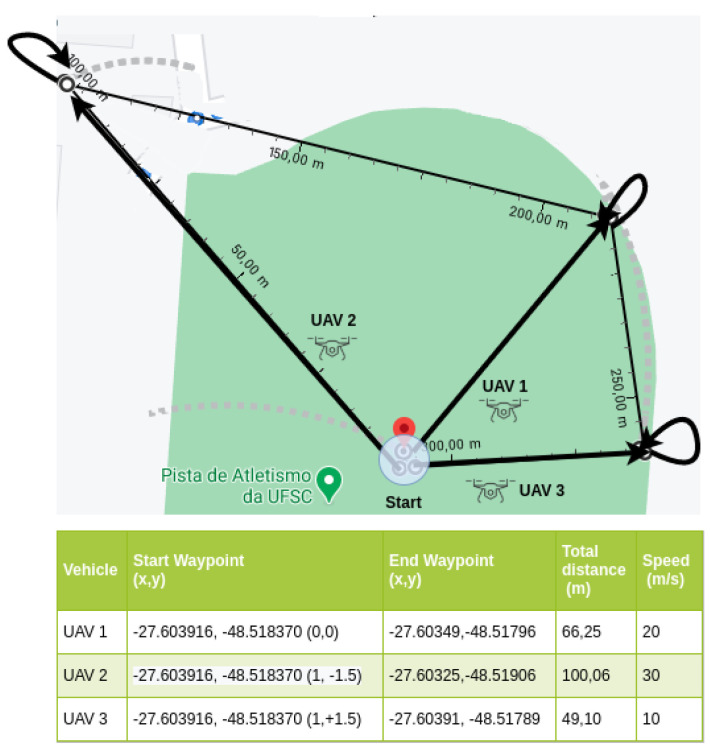
Experiment scenario-1 (3 UAV).

**Figure 5 sensors-21-04255-f005:**
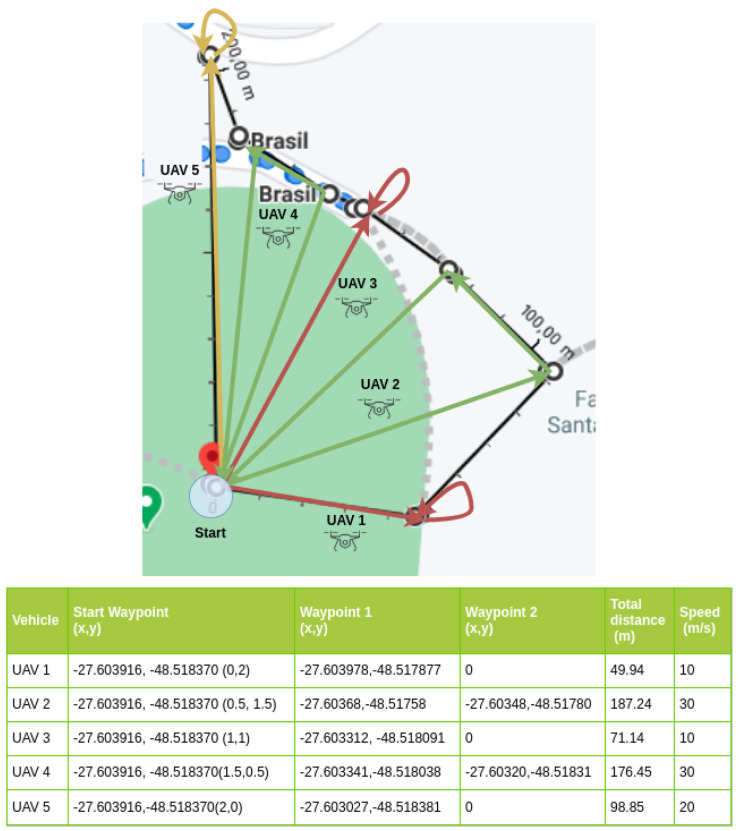
Experiment scenario-2 (5 UAV).

**Figure 6 sensors-21-04255-f006:**
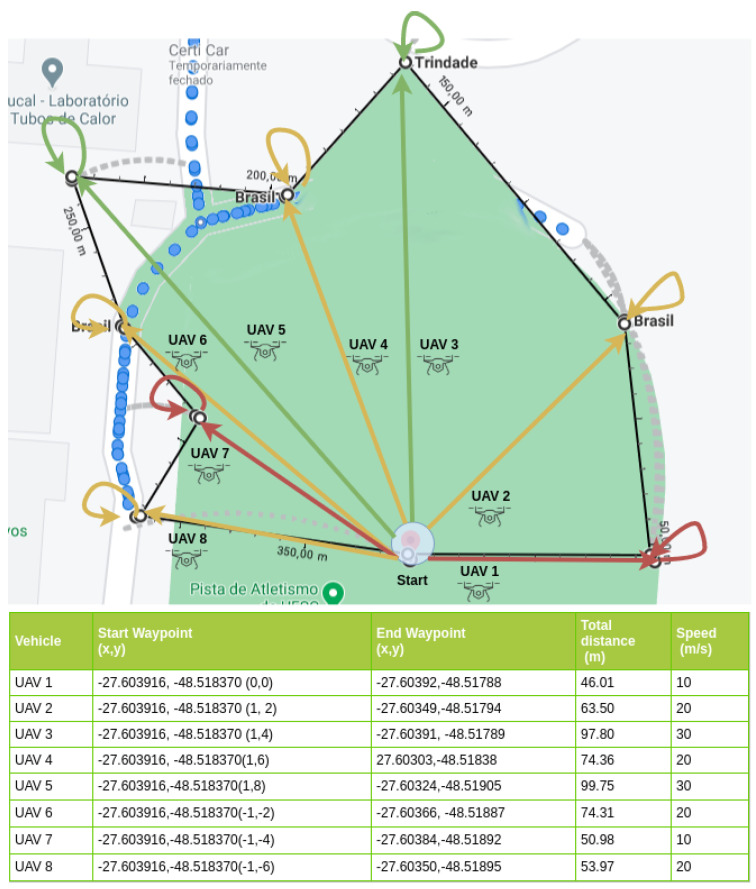
Experiment scenario-3 (8 UAV).

**Figure 7 sensors-21-04255-f007:**
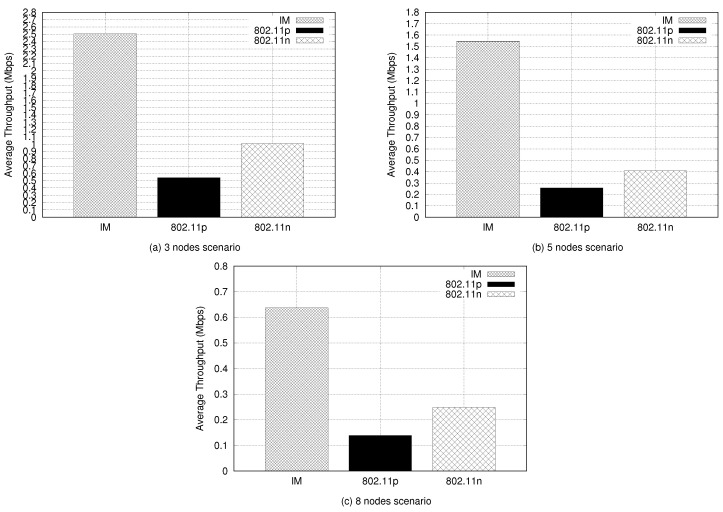
Average network throughput.

**Figure 8 sensors-21-04255-f008:**
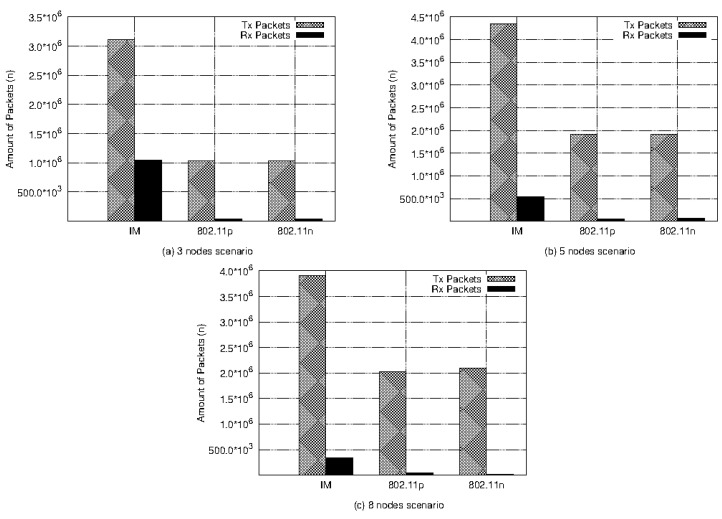
Amount of packets (*n*) trafficked.

**Figure 9 sensors-21-04255-f009:**
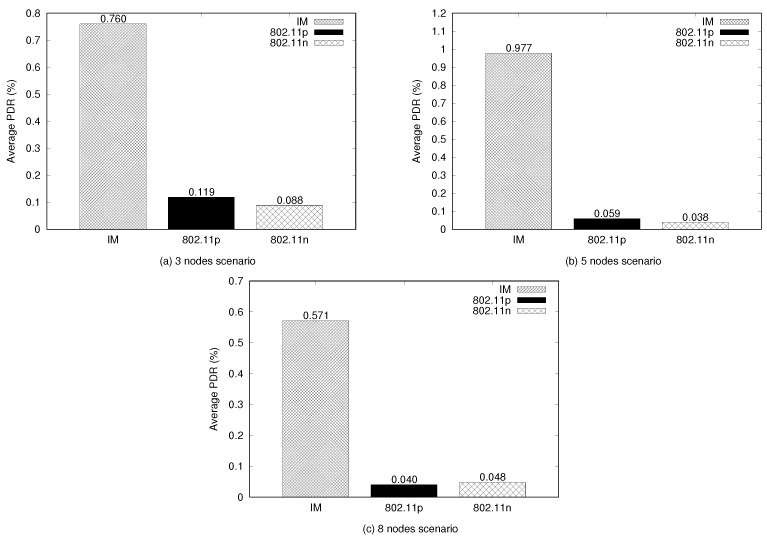
Packet delivery rate (%).

**Figure 10 sensors-21-04255-f010:**
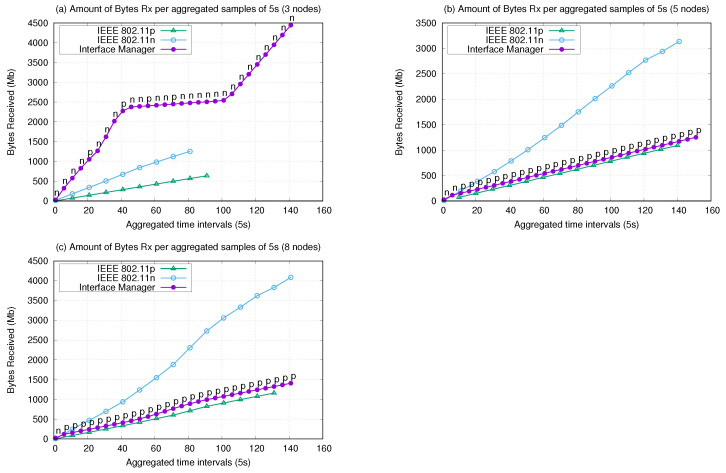
Bytes received (Mb) by the PHY layer by aggregated time intervals (s).

**Figure 11 sensors-21-04255-f011:**
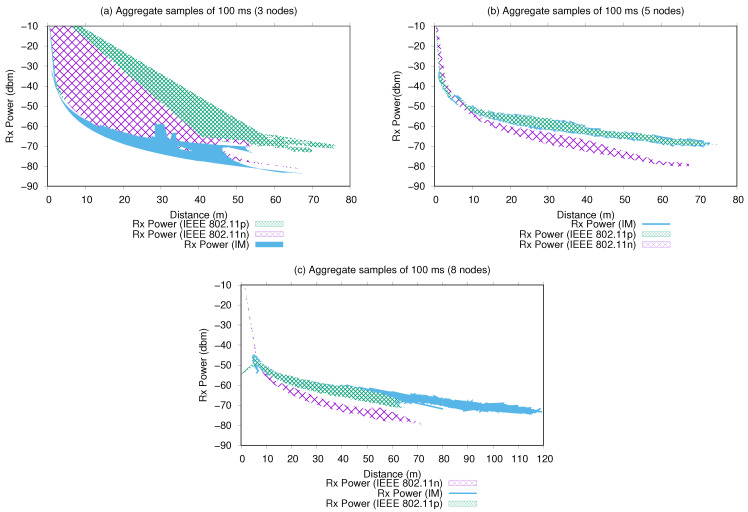
Rx power per distance (m) using aggregate samples intervals of 100 ms.

**Figure 12 sensors-21-04255-f012:**
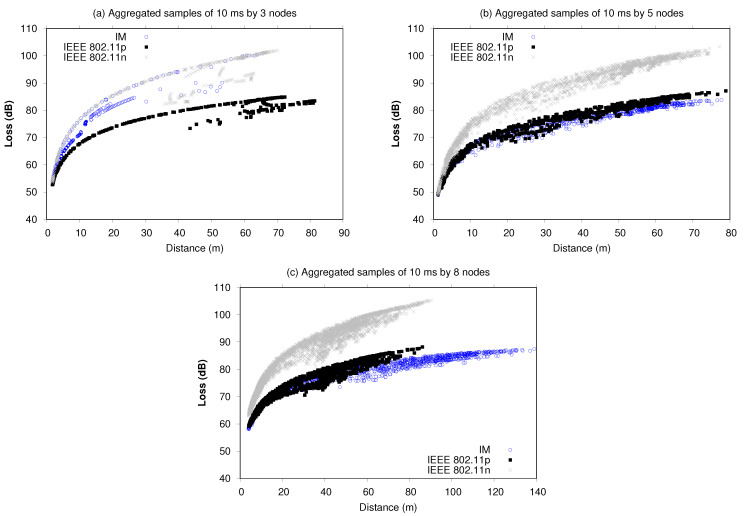
Loss (dB) versus distance (m) using aggregated samples of 10 ms.

**Figure 13 sensors-21-04255-f013:**
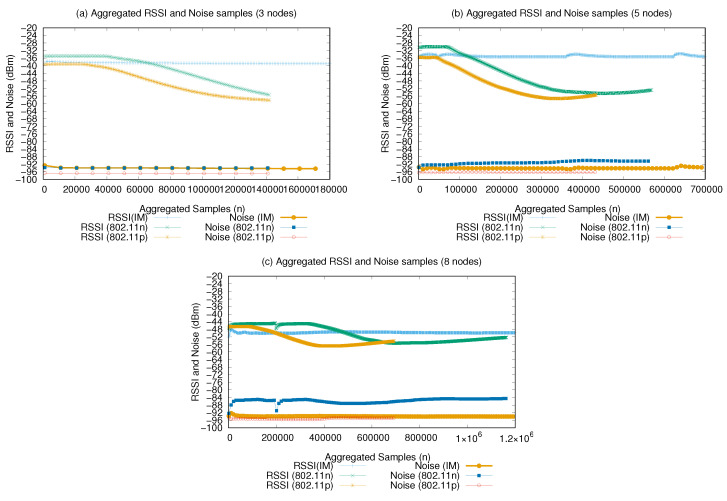
Aggregated RSSI and noise samples.

**Figure 14 sensors-21-04255-f014:**
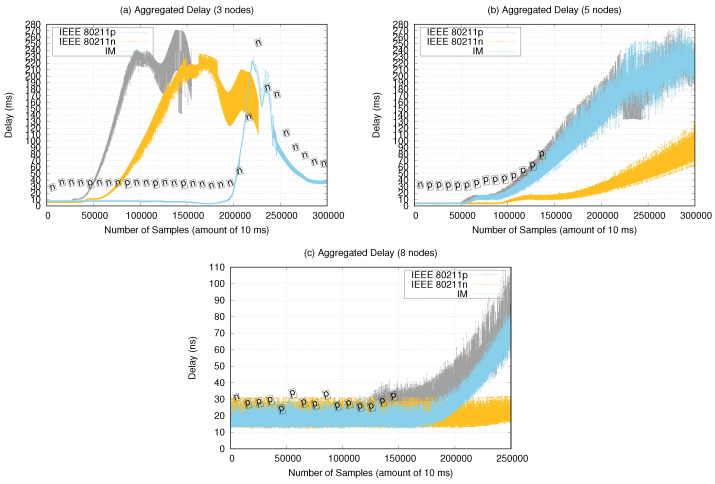
Delay (ms) aggregated by number of samples (amount of 10 ms).

**Table 1 sensors-21-04255-t001:** Metrics used as requirements for decision making.

Condition ($c_{i}$)	Network Metric
$c_{1}$	Throughput
$c_{2}$	Total Bytes Sent
$c_{3}$	Total Bytes Received
$c_{4}$	RSSI
$c_{5}$	SNR

**Table 2 sensors-21-04255-t002:** Settings adopted in the experiments.

Parameter	Setting
Network Topology	Ad Hoc Networks
Attenuation Model	Friis Free Space
Tx power	16.02 dBm (40 mW)
Maximum Speed	10, 20, and 30 m/s
Fading Model	Nakagami
Gazebo Duration Time	18 min (3), 1 h 30 min (5), 2 h (8)
Interface Manager Interval Decision	1 s
Sample Interval	10 ms,100 ms
Packet size	1472 bytes
Interface A	IEEE 80211n 2.4 GHz
Range Int. A	300 m
Channel Frequency Int. A	2.432 GHz (CH 5-6)
Interface B	IEEE 80211p 5 GHz
Range Int. B	350 m–1 km (theoretical)
Channel Frequency Int. B	5.860 GHz (CH 172)

**Table 3 sensors-21-04255-t003:** Average delay, latency, and flow of messages.

Experiment	Average End-to-End Delay (ms)	Latency (ms)	Flow (n)
3-node IM	2.34 × 10${}^{-2}$	29.112	45
3-node 802.11p	1.44 × 10${}^{-7}$	0.008	8
3-node 802.11n	4.50 × 10${}^{-4}$	28.835	8
5-node IM	5.98 × 10${}^{-1}$	2.000	78
5-node 802.11p	4.62 × 10${}^{-7}$	0.026	16
5-node 802.11n	6.50 × 10${}^{-4}$	38.979	16
8-node IM	2.37 × 10${}^{-1}$	2.137	123
8-node 802.11p	4.15 × 10${}^{-6}$	0.146	28
8-node 802.11n	1.02 × 10${}^{-3}$	38.210	28

## Data Availability

Not applicable.

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
