# Peer review of "Communication Interface Manager for Improving Performance of Heterogeneous UAV Networks"

_sensors, 2021, doi:10.3390/s21134255_

Round 1

Reviewer 1 Report

In this paper, the authors propose the Interface Manager (IM) which chooses the best communication interface by considering the metrics from MAC and PHY layers for improving message transmission rate and connection stability.

The aim of the proposed technique sounds reasonable and there are many simulation results.

However, the extensive editing of English language and presentation should be done, and many modifications should be required as follows:

  1. There are so many grammatical and presentation errors. It was so hard for me to understand the descriptions because of a lot of grammatical errors. There are many sentences which should be rephrased. There are many presentation errors which degrade the level of completion of the manuscript. One example is that “Figure ??” appears twice. The authors should follow the principles of academic presentation. For instance, the authors should use “do not” instead of “don’t”.
  2. The related work section (about 4 pages) accounts for too much of the manuscript’s total content.
  3. The authors should conduct the theoretical analysis to show the validity of the proposed decision scheme. Is it reasonable to simply add the metrics without weight factors? In addition, the authors should give a more detailed explanation about their system. Why did the proposed system choose to use IEEE 802.11n and 802.11p instead of other interfaces? Moreover, the author should present the clear definition of each metric they used in the manuscript.
  4. The Section 6, “Obtained Results”, should be improved. The many results are simply listed at length, so that it is hard for readers to get the key points of the results.

Author Response

Dear Reviewer 1,

First of all, we would like to thank you for providing such constructive reviews. We welcome all the valuable comments and recommendations that have been provided to us. We have found them all very useful in order to improve the quality of our work.

This new version of the manuscript has undergone English language editing by MDPI. The text has been checked for correct use of grammar and common technical terms, and edited to a level suitable for reporting research in a scholarly journal. 

Below, we provide our answers in reply to your comments from the original manuscript. Your comments are marked in green, and our answers in red. The performed changes in the paper itself are marked in blue.

1. There are so many grammatical and presentation errors. It was so hard for me to understand the descriptions because of a lot of grammatical errors. There are many sentences which should be rephrased. There are many presentation errors which degrade the level of completion of the manuscript. One example is that “Figure ??” appears twice. The authors should follow the principles of academic presentation. For instance, the authors should use “do not” instead of “don’t”.

We apologize for the many grammatical and presentation errors in the original manuscript. As previously mentioned, the paper has gone through professional English language editing. We hope it has improved readability accordingly. 

2. The related work section (about 4 pages) accounts for too much of the manuscript’s total content.

Indeed you are right. Section 2 (Related works) was schrinked in about one page by editing phrases and by removing references and related discussion that do not directly affect the comprehension of our proposal. Additional works were added in reply to reviewer’s 2 comments.

3a. The authors should conduct the theoretical analysis to show the validity of the proposed decision scheme. Is it reasonable to simply add the metrics without weight factors?

Thanks for the valuable feedback. Firstly, let me please add that given the extremely high dynamicity of the wireless medium, it becomes very difficult to predict in analytical terms what are the bounds of proposed Interface manager (IM) performance, regardless of the numbers of interfaces it might use. 

In the revised version of the manuscript we highlighted the theoretic maximum flow rate of each experimentation scenario, so that the reader can compare with such ideal but unrealistic bounds (lines 454 to 465). 

Besides, it is our understanding that weighting the current metrics might lead to worse decisions in the long run. 

3b. In addition, the authors should give a more detailed explanation about their system. Why did the proposed system choose to use IEEE 802.11n and 802.11p instead of other interfaces? 

A new paragraph was added to the introduction in order to better justify the choices for  IEEE 802.11n and 802.11p interfaces (lines 56 to 67). 

3c. Moreover, the author should present the clear definition of each metric they used in the manuscript.

Section 4 was edited to better clarify the adopted metrics and the variables used in the formulations. The equations were also revised. 

4. The Section 6, “Obtained Results”, should be improved. The many results are simply listed at length, so that it is hard for readers to get the key points of the results.

Section 6 (Obtained Results) was edited towards making the requested improvements, as follows:  

Text editing was conducted overall in order to better explain obtained results in both application level and MAC and PHY level. The aim was detailing to readers the advantages of switching between different communication interfaces for better adapting to the medium current conditions.  

An additional section named “Final remarks” (6.3) was added to highlight the advantages and drawbacks of the proposed IM. 

All graphs were changed to vector format in order to improve quality, allowing the reader to zoom in without quality loss. 

Reviewer 2 Report

Aiming to increase the stability of message delivery with link quality, this paper presents an Interface Manager (IM) that is capable of improving heterogeneous communication in multi-UAV networks.  

The method is fully evaluated by experiments and comparative studies. After carefully reading, I find that this paper is extremely interesting, it is very interesting and well organized. However, it has some problems that need to be solved further.

  1.  In related works, 2.1. "Homogeneous Systems" and 2.2 "Heterogenous Systems" are too long. It is suggested that the author simplify this part and highlight the description related to the article.
  2. Some Figures have poor visibility, such as Figure 1, Figure 2, and Figure 3. It is suggested that the authors modify them to color ones.
  3. Figure 7, 8, and 9 have several subgraphs and need to be represented separately, such as (a), (b) and (c).
  4. English language and style are fine/minor spell check required, such as Line 704.
  5. In addition, I suggest updating the background of more reference , which were published in the last three years.

In order to qualify for publication in Sensors, the paper must be improved according to the comments to the authors.

Author Response

Dear Reviewer 2,

First of all, we would like to thank you for providing such constructive reviews. We welcome all the valuable comments and recommendations that have been provided to us. We have found them all very useful in order to improve the quality of our work.

This new version of the manuscript has undergone English language editing by MDPI. The text has been checked for correct use of grammar and common technical terms, and edited to a level suitable for reporting research in a scholarly journal. 

Below, we provide our answers in reply to your comments from the original manuscript. Your comments are marked in green, and our answers in red. The performed changes in the paper itself are marked in blue.

The method is fully evaluated by experiments and comparative studies. After carefully reading, I find that this paper is extremely interesting, it is very interesting and well organized. However, it has some problems that need to be solved further.

Thank you very much for such positive feedback. 

 In related works, 2.1. "Homogeneous Systems" and 2.2 "Heterogenous Systems" are too long. It is suggested that the author simplify this part and highlight the description related to the article.

Indeed you are right. Section 2 (Related works) was schrinked in about one page by editing phrases and by removing references and related discussion that do not directly affect the comprehension of our proposal. 

Some Figures have poor visibility, such as Figure 1, Figure 2, and Figure 3. It is suggested that the authors modify them to color ones.

Figure 7, 8, and 9 have several subgraphs and need to be represented separately, such as (a), (b) and (c).

All graphs were changed to vector format in order to improve quality, allowing the reader to zoom in without quality loss. Special attention was given to those figures pointed out by this reviewer. 

English language and style are fine/minor spell check required, such as Line 704.

We apologize for the many grammatical and presentation errors in the original manuscript. As previously mentioned, the paper has gone through professional English language editing. We hope it has improved readability accordingly. 

In addition, I suggest updating the background of more reference , which were published in the last three years.

Additional relateds and up-to-date works were added in section 2: [25], [26], and [27]. 

Reviewer 3 Report

Dear Editor,

my comments as below:

  1. abstract, all acronyms must be expalined, e.g. UAVs, MAC, etc.
  2. chapter 2.1, please add more publications about application the GNSS system in UAV technology, especially about SBAS augmentation system in UAV.
  3. equation (1), what means cxi ?
  4. Figure 2 is illegible, please correct it.
  5. equation (3), what means h,ch,H?
  6. Figure 7 is illegible, please correct it.
  7. Figure 8 is illegible, please correct it.
  8. Figure 9 is illegible, please correct it.
  9. Figure 10 is illegible, please correct it.
  10. Figure 11 is illegible, please correct it.
  11. Figure 12 is illegible, please correct it.
  12. Figure 13 is illegible, please correct it.
  13. Figure 14 is illegible, please correct it.
  14. References, see point 2 in my comments.
  15. Abstract, please also underline your findings from research test, for example add the obtained results from research test.

In my opinion the paper is good, but my comments please take into account in your paper.

Author Response

Dear Reviewer 3,

First of all, we would like to thank you for providing such constructive reviews. We welcome all the valuable comments and recommendations that have been provided to us. We have found them all very useful in order to improve the quality of our work.

This new version of the manuscript has undergone English language editing by MDPI. The text has been checked for correct use of grammar and common technical terms, and edited to a level suitable for reporting research in a scholarly journal. 

Below, we provide our answers in reply to your comments from the original manuscript. Your comments are marked in green, and our answers in red. The performed changes in the paper itself are marked in blue.

abstract, all acronyms must be expalined, e.g. UAVs, MAC, etc.

Thank you for the feedback. It was edited accordingly. 

chapter 2.1, please add more publications about application the GNSS system in UAV technology, especially about SBAS augmentation system in UAV.

It was added in section 2.1, lines 166 to 197. 

equation (1), what means cxi ?

Done. See lines 319 to 322. 

Figure 2 is illegible, please correct it.

All graphs and figures were changed to vector format in order to improve quality, allowing the reader to zoom in without quality loss. 

equation (3), what means h,ch,H?

Done. See lines 360 to 365.

Figure 7 is illegible, please correct it.

Figure 8 is illegible, please correct it.

Figure 9 is illegible, please correct it.

Figure 10 is illegible, please correct it.

Figure 11 is illegible, please correct it.

Figure 12 is illegible, please correct it.

Figure 13 is illegible, please correct it.

Figure 14 is illegible, please correct it.

As mentioned above, all graphs and figures were changed to vector format in order to improve quality, allowing the reader to zoom in without quality loss. 

Abstract, please also underline your findings from research test, for example add the obtained results from research test.

Thank you for your comment. This was done in the last sentence of the revised abstract.

Round 2

Reviewer 1 Report

Overall, it seems that the authors worked hard to improve the quality of the paper. In particular, it had improved a lot in grammar and the many errors were eliminated, so the overall completeness was improved. However, there are still some improvements to be made as follows:

  1. In the answer of Q3a. the authors stated that “Besides, it is our understanding that weighting the current metrics might lead to worse decisions in the long run.”. The authors should explain why weighing the metrics lead to worse decisions in the long run, in the revised manuscript.
  2. The authors should examine the manuscript line by line, and the presentation can be improved more. There are still many points to be improved, but I list only a few as follows:
    1. Fly ad hoc networks -> Flying ad hoc networks (at line 28)
    2. Use a proper metric prefix. (e.g., 10,000,000 bps -> 10 Mbps in the line 459)
    3. Use a citation instead of inserting the Link into a sentence directly. (e.g., at line 465)

Author Response

Dear Reviewer 1,

We really appreciate your acknowledgement of our efforts. And we will be pleased to comment on the additional issues that you have addressed, as follows.

In the answer of Q3a. the authors stated that “Besides, it is our understanding that weighting the current metrics might lead to worse decisions in the long run.”. The authors should explain why weighing the metrics lead to worse decisions in the long run, in the revised manuscript.

Thanks for calling our attention to this topic. It is, indeed, an important aspect to be properly addressed in the paper. We have tackled it in this revised version of the manuscript in the following way:

1) We added the following paragraph in section 4 (lines 317 to 322):

Equal weights for each metric were adopted aiming to maintain connectivity and stability in the communication links, given that these metrics have the same importance in this kind of high-mobility wireless networks. So, it is not enough to simply have a larger volume of bytes received, but also how often these bytes are received and with which quality they are received. Otherwise, if in the presence of high loss and low RSSI, such received bytes could be constituted by trash or truncated messages.

2) We modified the last paragraph of section 6.3 (lines 851 to 856):

Finally, even with the lower flow rates presented by the IM (Figures 10 and 13), using the IM allowed increased stability in the transmissions, with continuous growth and less fluctuations, reducing the traffic interruptions caused by signal losses. This reinforces the assumption that the adopted metrics for selecting the best interface should be used with similar weights. If, for instance, a higher weight were given to the amount of bytes received, the 801.11n interface would have been selected, but there would not be as much successfully delivered messages at the destination nodes as achieved in the conducted experiments.

3) We modified the last paragraph of the paper (Future works directions) in lines 879 to 881:

In future work directions, the design of a new heuristic that considers different classes of communication messages should be addressed. This could, for instance, favor some of the metrics over the others. In Non-delay tolerant networks (NDTN), for example, it could be interesting to assign a higher weight for delay and throughput metrics. In this respect, our previous results in [14] will be considered regarding homogeneous network behavior (802.11n and 802.11p) using different message classes. 

The authors should examine the manuscript line by line, and the presentation can be improved more. There are still many points to be improved, but I list only a few as follows:

Fly ad hoc networks -> Flying ad hoc networks (at line 28)

Use a proper metric prefix. (e.g., 10,000,000 bps -> 10 Mbps in the line 459)

Use a citation instead of inserting the Link into a sentence directly. (e.g., at line 465)

Thanks again for calling our attention to such aspects. They were properly corrected, as well as a few other typos that were overlooked.

Reviewer 2 Report

Thank the authors for their efforts. The authors have adequately addressed all my concerns in the review, and did a good job to revise and improve the paper. The paper now is suitable for publication in Sensors in its current form.

Author Response

Thanks a lot for acknowledging our efforts and accepting our contribution.

Reviewer 3 Report

Dear Editor,

I accept the paper in current form.

Author Response

Thanks a lot for accepting our contribution.